# Patterns of healthcare services utilization associated with intimate partner violence (IPV): Effects of IPV screening and receiving information on support services in a cohort of perinatal women

**Nihaya Daoud**[1]*, **Lotan Kraun**[1,2], **Ruslan Sergienko**[1], **Naama Batat**[3], **Ilana Shoham-Vardi**[1], **Nadav Davidovitch**[2], **Arnon Cohen**[2,4]

**1** Department of Public Health, School of Public Health, Faculty of Health Sciences, Ben-Gurion University of the Negev, Beer Sheva, Israel, **2** Department of Health Systems Management, School of Public Health, Faculty of Health Sciences & Guilford Glazer Faculty of Business and Management, Ben-Gurion University of the Negev, Beer Sheva, Israel, **3** Clalit Health Services, Tel Aviv, Israel, **4** Siaal Research Center for Family Medicine and Primary Care, Faculty of Health Sciences, Ben-Gurion University of the Negev, Beer Sheva, Israel

* daoud@bgu.ac.il

**Data Availability Statement:** The data that support the findings of this study are available from Clalit

## Abstract

### Background

While women experiencing intimate partner violence (IPV) face significant health consequences, their patterns of healthcare services (HCS) utilization are unclear, as are the effects of IPV screening and receiving information on these patterns.

### Objectives

1. Compare utilization patterns of five HCS (visits to family physician, gynecologist, specialist and emergency room, and hospitalization) in a cohort of perinatal women who reported experiencing versus not experiencing any IPV and IPV types (physical and/or sexual; emotional and/or verbal; social and economic); 2. Examine whether IPV screening, receiving information on support services, or both, affect patterns; and 3. Compare these associations between ethnic groups (Arab and Jewish women).

### Methods

We conducted a prospective study using registry data on HCS utilization obtained from Israel's largest Health Fund (Clalit) in the year following a 2014–2015 survey of a cohort of 868 perinatal women in Israel (327 Arab minority, 542 Jewish) on their reports of experiencing IPV, IPV screening, and receiving information. Using multivariate analysis, we calculated adjusted odds ratios (AOR) and 95% confidence intervals (CI) for the five HCS utilizations in association with reports of any IPV and IPV types. We adjusted for IPV screening, receiving information about services, and both, in the total sample, and separately among ethnic groups.

Health Services, but restrictions apply. These data were used under license for the current study, and so are not publicly available. They are, however, available on reasonable request from Prof. Arnon Cohen, Chief Physician's Office, Clalit Health Services, Arlozorov St 115, Tel Aviv-Yafo, 6209813, Israel: arcohen@clalit.org.il

**Funding:** This follow up study was funded by The Israel National Institute of Health Policy and Health Research. http://www.israelhpr.org.il/e/. Dr. Nihaya Daoud received a grant (# R/22/2105). The original study was funded by the Israel Science Foundation https://www.isf.org.il/#/ received by Dr. Nihaya Daoud and Ilana Shoham-Vardi (Grant # 881/13). The funders had no role in the study design, data collection and analysis, decision to publish, or preparation of the manuscript.

**Competing interests:** The authors have declared that no competing interests exist.

**Abbreviations:** VAW, Violence against women; IPV, intimate partner violence; HCP, Healthcare providers; HCS, Healthcare services; ER, Emergency room; MOH, Ministry of Health; MCH, Maternal and Child Health.

## Results

Any IPV and IPV types had significant associations with some HCS utilization variables, with different directions and patterns for the ethnic groups. Experiencing IPV was associated with higher HCS utilization among Arab women, lower utilization in Jewish women. Arab women experiencing IPV were twice as likely to visit a gynecologist than women not experiencing IPV (AOR (95% CI) was 2.00, 1.14–3.51 for any IPV; 2.17, 1.23–3.81 for emotional and/or verbal IPV, and 1.83, 1.04–3.22, for social and economic IPV). Among Jewish women, experiencing any IPV was associated with lower likelihood of emergency-room visits (0.62, 0.41–0.93); and experiencing physical and/or sexual IPV was associated with lower likelihood of family physician visits (OR = 0.20, 0.05–0.82). Both IPV screening and receiving information were associated with lower HCS utilization among Arab women only.

## Conclusions

Different HCS utilization patterns among women who reported experiencing versus not experiencing IPV in different ethnic groups suggest complex relationships that hinge on how HCS address women's needs, starting with IPV screening and providing information. This might inform tailored programs to tackle IPV at the HCS, particularly for minority women.

## Introduction

About one third of women globally face intimate partner violence (IPV), resulting in severe harm to their health [1, 2]. IPV refers to any behavior within an intimate relationship that causes harm to those in the relationship. This includes physical, psychological, sexual, economic, or social abuse, as well as controlling behaviors [3, 4]. Women experience more IPV compared to men [5]. As a result they are more likely to suffer the consequences of IPV; injury, sexually transmitted infections and HIV; depression, sleep and eating disorders; and alcohol and drug addictions [1, 2, 6]. Women who experience IPV also have more emotional distress and suicide attempts [7], as well as anxiety and postpartum depression [1]. In addition, IPV can lead to sexual dysfunction, unwanted pregnancy, and unsafe abortion [8, 9]. As a result, women who experience IPV tend to seek more help through health care services (HCS) compared to women not experiencing IPV [10, 11], in some cases twice as much [12–15], with higher-than-average visits to a primary-care physicians, specialists and pharmacies [15]. An European Union (EU) study on violence against women (VAW) [16] found that 26% of women experiencing physical and sexual IPV, and 34% of women experiencing sexual violence used HCS [16]. In Spain [17] a longer period of experiencing physical and mental violence predicted increased HCS use [17]. Generally, medical care costs for a woman with experience of IPV were 1.6–2.3 times greater compared to women had not experienced IPV [18]. Emergency-room (ER) visits costs were higher, hospitalization costs were twice as high [13, 19], and overall, medical care costs were three times higher among women experiencing IPV compared to women not experiencing IPV [14].

### Complex association between experiencing IPV and HCS use

Despite these findings, other studies paint a more ambivalent picture of HCS utilization among women experiencing IPV [20]. Such studies suggest that many of these women face barriers limiting their ability to seek help via HCS, and that they might use these services less

than they need to [21–24]. For example, they might delay visits to family physicians and medical treatment in order to avoid disclosing IPV [25]. Abusive partners also try to block women from receiving medical care [25]. Other issues adding to women's reluctance to get professional support services for IPV through HCS include their preference to get this help from loved ones, the belief that IPV is not a serious problem, lack of trust in HCS, ignorance about where to get HCS help, lack of access to HCS, financial constraints, shame, confidentiality concerns, and fear of being blamed [16]. Therefore, a complex set of factors might together create barriers to HCS utilization, depending on the type and organizational structure of the healthcare system (universal versus private HCS, fees, etc.) and its handling of VAW [26]. External societal factors surrounding women's status and the actions of perpetrators [20] can also affect women's help seeking behaviors and whether women disclose IPV to a HCP [27, 28]. Meanwhile, a related gap remains understudied among minority women experiencing IPV, whose help seeking was even lower [29, 30], as they face more social and economic barriers to HCS use than others [23, 31–34].

## IPV screening policy and counseling for IPV services

To encourage women experiencing IPV to disclose abuse and seek help through HCS, many countries have policies to ensure IPV screening and provision of information regarding relevant support services [35]. IPV screening can take various forms, including universal screening; selective or targeted screening; and routine inquiry [35]. Each strategy specifies which characteristics to look for when women present at HCS, and which tools to use for screening, resulting in varying coverage [35, 36]. Although evidence on the effectiveness of IPV screening in HCS is lacking [37], many medical organizations [38–42], including the WHO [43], support some form of screening or case finding for IPV, as well as information provision and referral. Yet, overall, HCPs are reluctant to conduct any form of IPV screening [41, 44, 45], as they too face barriers, including time shortages, lack of training, and adverse personal attitudes, while lacking system-level guidelines about how to screen for IPV [41, 44–46]. A recent systematic review focused on screening and counseling within HCS for women experiencing IPV found that HCPs need such system-level support in order to overcome barriers for providing these services [46]. This should include not just training, but also clear guidance and strategies for screening and counseling policies [46]. Improving these services in the HCS might be effective not only in providing safety planning for women, but could affect their HCS utilization and reduce costs. However, no study we know of has focused on the effects of IPV screening and information provision on HCS utilization patterns among women experiencing IPV compared to those not experiencing IPV.

## HCS utilization, screening, and receiving information among women experiencing IPV in Israel

In Israel, no research has considered patterns of HCS utilization among women experiencing IPV. Annually, about 5,000 women experiencing violence seek help through HCS [47]. Most experience sexual violence (47%), followed by emotional violence (18%) [47]. In 2015, among these women, about 45% visited a hospital, almost 35% visited a community-based healthcare clinic, and the rest visited maternal and child health (MCH) clinics [47]. In the same period, familial violence accounted for about 20,000 police reports; Jewish women filed 70% of these, Arab women the rest [48]. Our recent research showed total IPV prevalence of 38.9% in Israel (4.6% physical and/or sexual, 28.6% emotional and/or verbal, and nearly 26.1% social and economic). The proportion of any IPV and types of IPV was higher among Arab women and women with low socioeconomic status [49].

In response to violence against women and IPV, Israel's Ministry of Health (MOH) has launched several initiatives [50], including a policy to ensure IPV screening, information provision, and referrals through HCS [50]. The MOH instructed HCS organizations (health funds), hospitals, and public health services to establish committees for handling VAW and IPV [50]. Nonetheless, a quality assurance report on treatment of VAW at HCS [51] identified barriers faced by healthcare providers when attempting to respond to VAW, including HCS organizational structures, as well as personal and societal barriers [51]. Personal barriers related to HCPs' attitudes and perceptions regarding IPV, while societal barriers were system-level barriers limiting access to HCS and social services organized to provide professional support to women experiencing IPV [51]. As well, another recent study showed that only half of women experiencing IPV reported ever being asked at HCS about experiencing IPV or receiving information about support services [52]. Indeed, women at higher risk for IPV reported being less frequently screened or receiving information at HCS [52]. Further, compared to Jewish women, Arab women reported being less often screened for IPV and receiving information [52].

Generally, despite provision of universal HCS since 1994 [53] and equality of access to care under law, utilization of, and access to HCS in Israel still differs dramatically between Arab and Jewish citizens [54, 55]. For example, Arab minority citizens use less specialist care [55] and cardiac prevention rehabilitation programs [56]. Arabs also live in peripheral areas in Israel's southern and northern regions, where some services are less available or accessible [57, 58]. Many mental health services are lacking in Arab villages and towns [57]. Indeed, Arabs use less mental health services, despite higher prevalence of mental health problems among them [58, 59]. In addition, Arab women face cultural barriers in accessing some HCS, such as family physicians [60]. This suggests different patterns of HCS utilization among Arab and Jewish women in general.

Regarding IPV services, previous research has identified many barriers faced by Arab women experiencing IPV in getting help through social and mental HCS [61], including a belief that IPV is a familial problem. Disclosure, including to healthcare providers, might put Arab women experiencing IPV, their families, and their children in jeopardy [61, 62]. Arab women are also reluctant to seek police assistance, as many view police as instruments of Arab oppression in Israel [61, 62].

## Study aims

The aims of this study are threefold: first, to examine the associations between any IPV and types of IPV with utilization of five medical HCS in a cohort of eligible perinatal women who participated in a 2014–2015 study on "Family Relations, Violence and Health" [49]; second, to examine the effect of 'ever been screened' for IPV and 'received information' about support services on the associations between IPV, IPV types, and HCS utilization; third, to compare these associations among Arab minority and Jewish majority women in Israel.

We hypothesized that women who reported experiencing any IPV or different types of IPV would utilize more HCS, and that IPV screening and receiving information would decrease medical HCS utilization among women experiencing IPV (compared to women not experiencing IPV), as women who are screened for, and receive information on IPV support services would be referred to specialized social and mental HCS. Finally, we hypothesized that these associations would differ by ethno-national group, with Arab minority women who experience IPV using fewer HCS than Jewish women who experience IPV, as Arab women face comparatively more barriers in accessing HCS [60].

## Methods

### Study design and data

This prospective study is based on linkage of registry data obtained from a large Israeli health fund (Clalit Health Services) on the use of five medical HCS, to survey data that we collected in our study on "Family Relations, Violence and Health" about IPV, types of IPV, IPV screening, and receiving information, as reported by the women we surveyed [63]. That original study consisted of a multi-layered cluster sample of 1,401 Arab and Jewish women aged 16–48 who were interviewed during their visits to 63 MCH Clinics in five districts of Israel's MOH in 2014–15. First, we randomly sampled MCH, with each clinic forming a cluster based on the geographical area and the district (Beersheba, Ashkelon, Central, Haifa and Nazareth). The number of MCH clinics in the various districts was calculated proportionally to the number of births and women's ethnicities (Arab and Jewish). We ended up with 63 MCH clinics: 21 in Arab-speaking neighborhoods, 33 in Hebrew-speaking neighborhoods, 9 in mixed cities.

### Data collection in the original study

Data on any IPV and types of IPV were collected from April 2014 to April 2015 in the MCH clinics. Study coordinators distributed informational leaflets at the sampled clinics. Then trained interviewers asked eligible mothers attending these MCH clinics (pregnant women, or those 6 weeks to 6 months postpartum, who speak Arabic or Hebrew) if they would participate in the study. Women who agreed and met the inclusion criteria signed an informed consent form and were interviewed face to face in their mother tongue (Arabic or Hebrew) in a private room using a structured questionnaire. Women were then asked to sign a second informed consent form to obtain data on their use of HCS (hospital care and community health services). Interviewers were instructed to keep the completed questionnaires in a closed envelope at the MCH clinic head nurse's office. Informed consent forms were kept in a separate envelope from the questionnaires. The research coordinator collected the envelopes once monthly. Rate of response among Arab women was 76%, while among Jewish women it was 73%.

### Data linkage

Data on utilization of HCS was linked to the study data by the Clalit Health Services team using a code identifier. Among the 1401women, 869 were identified as members of Clalit Health Services based to their ID number. The other women were not members of Clalit and were excluded from the current analysis. As the Health Fund has information about IPV and IPV screening, our team sent Clalit a file with the women's ID number, and they returned an encrypted file that included each woman's code identifier and data on the use of HCS. The data file on all 869 women was stored on computers of the Faculty of Health Sciences at Ben-Gurion University, with access strictly limited to some research team members.

### Study sample and power calculation

In the current analysis we included only women from the original sample who were insured by Clalit Health Services between 2014 and 2018. Exclusion criteria were: lack of identifying details (i.e., ID card). Of the total sample of 1401 women who participated in the original study, 869 women were insured by Clalit and included in the current analysis (65% of participants in the original study).

Statistics power ranged from 84% for testing the association between IPV and HCS utilization in the community, to 97% for tertiary (hospital) HCS utilization. Calculations of power were conducted using WINPEPI [64] and were based on experiences of any IPV found in the

original study[49], and an odds ratio of 2.3 for HCS utilization among women experiencing IPV compared to those not experiencing IPV (based on findings of a previous US study) [18].

## Ethics approval

The original study was approved by the Ethics Committee of Ben-Gurion University of the Negev (# 1128–1) and by the MOH's Director of Public Health Services. The current study on IPV and HCS utilization was approved by the Helsinki committee of Clalit Health Services (# 0157-15-COM2). Study measures

**Healthcare services (HCS) utilization** was measured based on use of five medical services. Three services were community-based: visits to family physician, to a gynecologist, and to another specialist. The other two were within the framework of hospital care: ER visits and number of hospitalizations in surgical and internal medicine departments. Prospective data on these variables were obtained from Clalit Health Services for 24 months of follow-up, starting from the original interview date. In the second year, there was a small number of HCS utilizations, however, so we made the decision to use data from the first year after the interview only (12 months). After examining the distribution of HCS use variables, we dichotomized data into yes (used the services) or no (did not use the service) by the median score (= 0) for four of five variables. For the fifth variable, related to visits to family physicians, we dichotomized data as follows: 1. low use (0–4 visits for a period of one year)) as we estimated one visit per 3 months and 2. high use (5 or more visits during a period of one year).

As for the independent variables, data were obtained from the original survey [49] and included: **Any intimate partner violence (IPV):** self-reported experiences of any 10 acts of violence perpetrated at any time by a woman's intimate partner [49, 65]. These included, for example: 'Your partner has hit you, kicked you, pushed you, or thrown things at you?'; 'Are you fearful of drastic changes in your partner's mood?', and 'Does your partner try to isolate you from your family and friends?' If a woman reported any of these, she was assigned as experiencing IPV for the purposes of this study. [49]. **Types of IPV:** experiences of one of the three types of IPV emerging from factor analysis for the list of 10 acts of any IPV: physical and/or sexual violence; emotional and/or verbal violence; and social and economic violence [49]. **IPV screening:** measured by a direct yes/ no question on whether or not anyone in HCS ever asked a woman if she experienced IPV. **Received information (RI) about IPV supportive services:** women were asked if they ever received information about what to do in case they experience IPV (regardless of responses to the IPV screening question). **Both screened and received information**: an index variable created by the sum of positive answers (yes screened, and yes received information) to the previous two questions versus all other answers [52].

**Ethnicity** was self-determined (Arab or Jewish). **Age** included three categories: 16–24, 25–34 and 35–44 years old. **Marital status** was categorized as married or not married (including single, divorced, separated, not-cohabitating, or other). Pregnancy status during the interview was determined by a direct question: "Are you currently pregnant or after birth?" **Mother of children** was determined with a direct yes/ no question: 'Do you have children?' **Women's education** we categorized into: 1. High school or less, 2. Postsecondary education, and 3. University education (Bachelor's, Master's, or Doctorate). **Chronic Disease** was measured by a direct yes/ no question: "Has the doctor ever told you that you have any chronic disease?"

## Statistical analysis

We conducted data analysis using SPSS version 25 [66]. Missing values were less than 5% for all study variables. The level of significance in the study was set to 10% or less. After data

cleaning and examining the distribution of all study variables, we conducted univariate analysis for the associations between any IPV and IPV types (physical and/or sexual; emotional and/or verbal; and social and economic) and utilization of medical HCS (family physician, gynecologist, specialist, ER, and hospitalization) for the total sample and for each ethnic group (Arab and Jewish). We used Chi Square test or t-test for the univariate analysis. We then conducted multivariable logistic regressions using generalized estimation equation (GENLIN procedure), as the study included a stratified cluster sample of MCH clinics. The multivariable analysis was conducted separately for each of the five HCS utilization variables in association with IPV and types of IPV, and included adjusted models for women's age, pregnancy status, education, and chronic illness, as these characteristics had significant univariate associations with at least one of the HCS utilization variables and with any IPV and types of IPV. The variables 'marital status' and 'mother of children' were not included in the multivariable models, as there were very few unmarried persons in the Arab women's study group, and few Jewish women had no children, which affected the convergence of the models.

Since we were interested in the effect of IPV screening, receiving information and both (being screened and received information) on the associations between IPV variables and HCS utilization in each study group, we conducted 5 more logistic regression models with generalized estimation equation (GENLIN) procedure for each of the five HCS utilization variables separately among Arab and Jewish women. The models were as follows: Model 1 was unadjusted, and Model 2 was adjusted for the socio-demographic variables (age, women's education, pregnancy status, and chronic disease). In addition to adjustments in Models 2, the next models adjusted for IPV screening (Model 3), received information (Model 4), and both (Model 5).

## Results

Women's mean age was 29 (SD = 5.3, range 17–44), with about 60% between 25 and 34 years old. Most were married, about one quarter were pregnant at the time of the interview, and the rest were up to six months after birth. Most women had children. More than half had postsecondary college or university education. Chronic illness was reported by 9.7% of the women (Table 1).

We observed significant differences between Arab and Jewish women with regards to socio-demographic characteristics (Table 1). Compared to Jewish women, Arab women were significantly younger, more often married, more often pregnant at the time of the interview, and had lower education. More Jewish women were mothers of children, and more Jewish women reported chronic illness (Table 1).

Regarding utilization of HCS, Table 2 shows that, for the total sample, visiting a gynecologist was the highest type of use (close to 75%), family physician visits were second (about 55% had high use), then visits to a specialist (48.4%), then ER (31.6%), and finally hospitalization (9.8%).

Table 2 also shows no significant differences between Arab and Jewish women regarding the use of a family physician and gynecologist services. However, we found significant differences between Arab and Jewish women regarding use of specialist services. Jewish women used this service more than Arab women (53.3% vs. 40.4%, respectively). Arab women visited the ER more (36.1% vs. 29.0%, respectively) and were more often hospitalized (13.5% vs. 7.6%, respectively) compared to Jewish women (Table 2).

Another important result concerned women's experiences of IPV. Fig 1 shows that, of the total sample, 41.8% of women reported experiencing any IPV, 30.7% experienced emotional and/or verbal IPV, 28.3% experienced social and economic IPV, and 5.4% physical and/or

**Table 1. Socio-demographic, socioeconomic, and health characteristics of the study sample by ethnic group (Jewish and Arab women).**

| | Total sample N = 869 | Jewish women N = 542 (62.4%) | Arab women N = 327 (37.6%) | P |
|---|---|---|---|---|
| | N (%) | N (%) | N (%) | |
| **Age** | | | | < 0.001 |
| 16–24 | 174 (20.0) | 43 (7.9) | 131 (40.1) | |
| 25–34 | 520 (59.0) | 360 (66.4) | 160 (48.9) | |
| 35–48 | 175 (20.1) | 139 (25.6) | 36 (11.0) | |
| **Marital status** | | | | < 0.001 |
| Married | 832 (96.1) | 508 (94.2) | 324 (99.1) | |
| Other | 34 (3.9) | 31 (5.8) | 3 (0.9) | |
| **Woman's status** | | | | < 0.001 |
| Pregnant | 202 (23.4) | 79 (14.6) | 123 (38.0) | |
| After birth | 662 (76.1) | 461 (85.4) | 201 (62.0) | |
| **Mother of children** | | | | < 0.001 |
| Yes | 808 (93.1) | 520 (96.1) | 288 (88.1) | |
| No | 60 (6.9) | 21 (3.9) | 39 (11.9) | |
| **Education (woman)** | | | | < 0.001 |
| High school or less | 375 (43.2) | 171 (31.5) | 204 (62.4) | |
| Postsecondary or college | 139 (16.0) | 91 (16.8) | 48 (14.7) | |
| Bachelor's or above | 355 (40.9) | 280 (51.7) | 75 (22.9) | |
| **Chronic disease** | | | | |
| Yes | 83 (9.7) | 76 (14.5) | 7 (2.1) | < 0.001 |
| No | 769 (89.7) | 449 (85.5) | 320 (97.9) | |

sexual IPV. Arab women reported significantly higher any IPV than Jewish women (66.7% vs. 26.8%), and this was true for all types of IPV (Fig 1).

Fig 2 presents the distribution of results concerning women's reports of 'ever been screened' and 'ever received information on support services for women experiencing IPV' or both (ever screened and received information). In the total sample, close to half of women (48.8%) reported ever being screened, and close to this number (47.2%) reported having received information about support services. Almost one third reported receiving both. However, significant differences emerged between Arab and Jewish women, with Arab women less often reporting ever being screened, receiving information, or both.

## Associations between any IPV, types of IPV, and utilization of HCS

Table 3 presents the univariate associations between any IPV, IPV types, and each of the five types of HCS utilization for the total sample and for each study group (Arab and Jewish women). For the total sample (Table 3A), we found significant associations (P<0.01) between IPV and HCS utilization. However, the direction of the association was not consistent. For

**Table 2. —HCS utilization rates in the total sample and among Jewish and Arab women during the first year after the interview.**

| | Family physician | Gynecologist | Specialist | ER | Hospitalization |
|---|---|---|---|---|---|
| | N (%) | N (%) | N (%) | N (%) | N (%) |
| *P-value* | *0.337* | *0.501* | *0.001* | *0.018* | *0.004* |
| **Jewish women** (N = 542) | 301 (55.5) | 404 (74.5) | 289 (53.3) | 157 (29.0) | 41 (7.6) |
| **Arab women** (N = 327) | 176 (53.8) | 243 (74.3) | 132 (40.4) | 118 (36.1) | 44 (13.5) |
| **Total Sample** (N = 869) | 477 (54.9) | 647 (74.5) | 421 (48.4) | 275 (31.6) | 85 (9.8) |

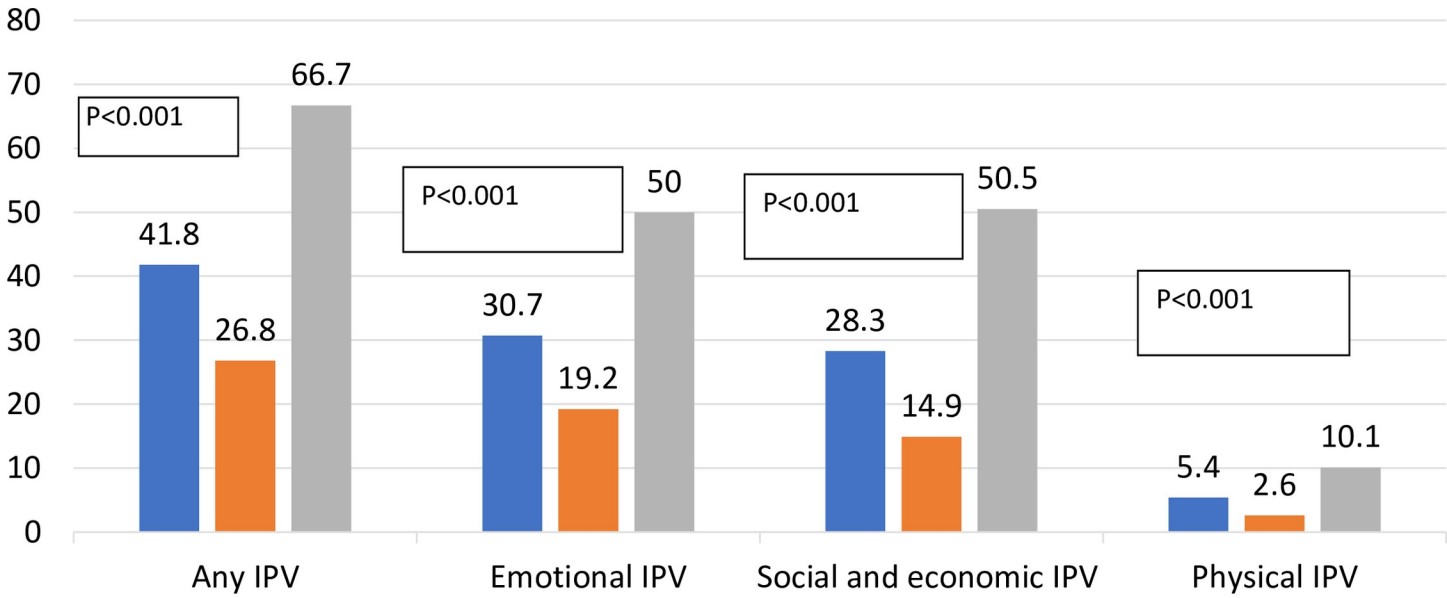

**Fig 1. Experiencing IPV (%) in the total sample and among Arab and Jewish women.**

example, use of family physician services was higher among women who reported not experiencing physical and/or sexual and emotional and/or verbal IPV. Similarly, higher use of a specialist services was associated with not experiencing any IPV, emotional and/or verbal, or social and economic IPV. However, using gynecologist services and hospitalization were higher among women who experienced any IPV, emotional and/or verbal, and social and economic IPV, compared to women who reported not experiencing these types of IPV. Visiting

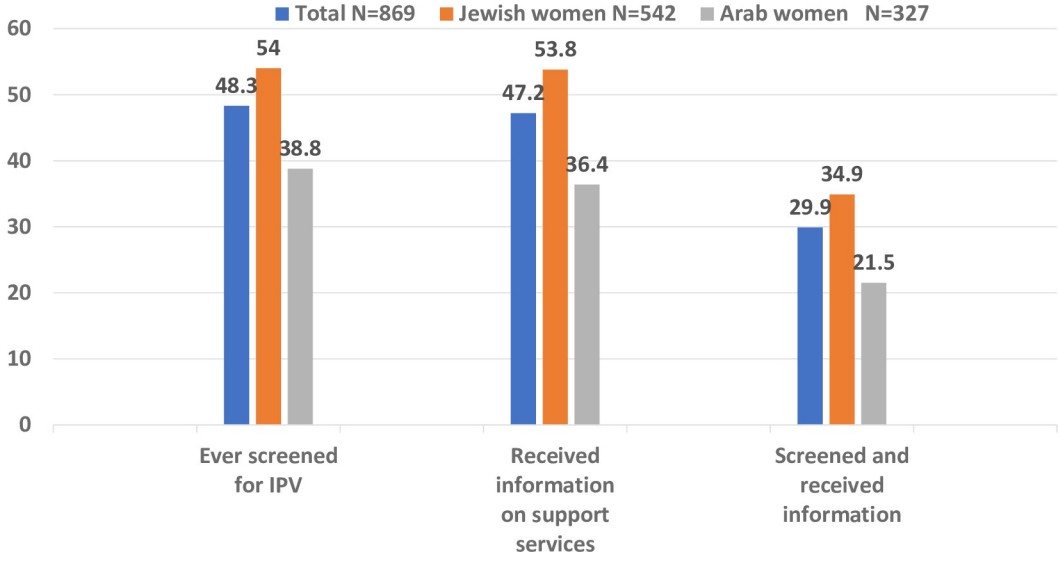

**Fig 2. Screening for IPV, receiving information about support services, and both, in the study groups.**

**Table 3. Univariate associations between any IPV, IPV types, and HCS utilization in the total sample and study groups.**

| | | Family physician | | Gynecologist | | Specialist | | ER | | Hospitalization | |
|---|---|---|---|---|---|---|---|---|---|---|---|
| IPV | Total N (%) | High use N (%) | P | Yes N (%) | P | Yes N (%) | P | Yes N (%) | P | Yes N (%) | P |
| | | | | | | 3a- Total sample—First year (N = 869) | | | | | |
| **Any IPV** | | | 0.404 | | **0.053** | | **0.033** | | 0.203 | | **0.011** |
| Yes | 363 (41.8) | 197 (54.3) | | 281 (77.4) | | 162 (44.6) | | 121 (33.3) | | 46 (12.7) | |
| No | 506 (58.2) | 280 (55.3) | | 366 (72.3) | | 259 (51.2) | | 154 (30.4) | | 39 (7.7) | |
| **Physical and/or sexual IPV** | | | **0.055** | | 0.556 | | 0.167 | | 0.33 | | 0.497 |
| Yes | 47 (5.4) | 20 (42.6) | | 35 (74.5) | | 19 (40.4) | | 13 (27.7) | | 5 (10.6) | |
| No | 820 (94.6) | 456 (55.6) | | 611 (74.5) | | 400 (48.8) | | 262 (32.0) | | 80 (9.8) | |
| **Emotional and/or verbal IPV** | | | **0.071** | | **0.024** | | **0.022** | | 0.112 | | **0.003** |
| Yes | 267 (30.8) | 136 (50.9) | | 211 (79.0) | | 115 (43.1) | | 93 (34.8) | | 38 (14.2) | |
| No | 599 (69.2) | 339 (56.6) | | 434 (72.5) | | 304 (50.8) | | 182 (30.4) | | 47 (7.8) | |
| **Social and economic IPV** | | | 0.446 | | **0.038** | | **0.010** | | 0.147 | | **0.033** |
| Yes | 246 (28.4) | 342 (55.1) | | 194 (78.9) | | 103 (41.9) | | 85 (34.6) | | 32 (13.0) | |
| No | 621 (71.6) | 134 (54.5) | | 452 (72.8) | | 316 (50.9) | | 190 (30.6) | | 53 (8.5) | |
| | | | | | | 3b- Jewish women (N = 542) | | | | | |
| **Any IPV** | | | 0.497 | | 0.445 | | 0.292 | | **0.053** | | 0.414 |
| Yes | 145 (26.8) | 80 (55.2) | | 107 (73.8) | | 74 (51.0) | | 34 (23.4) | | 12 (8.3) | |
| No | 397 (73.2) | 221 (55.7) | | 297 (74.8) | | 215 (54.2) | | 123 (31.0) | | 29 (7.3) | |
| **Physical and/or sexual IPV** | | | **0.009** | | 0.267 | | 0.304 | | **0.053** | | 0.326 |
| Yes | 14 (2.6) | 3 (21.4) | | 12 (85.7) | | 6 (42.9) | | 1 (7.1) | | 0 (0.0) | |
| No | 526 (97.4) | 297 (56.5) | | 391 (74.3) | | 281 (53.4) | | 156 (29.7) | | 41 (7.8) | |
| **Emotional and/or verbal IPV** | | | 0.174 | | 0.417 | | 0.481 | | 0.127 | | 0.389 |
| Yes | 104 (19.3) | 53 (51.0) | | 79 (76.0) | | 56 (53.8) | | 25 (24.0) | | 9 (8.7) | |
| No | 436 (80.7) | 247 (56.7) | | 324 (74.3) | | 231 (53.0) | | 132 (30.3) | | 32 (7.3) | |
| **Social and economic IPV** | | | 0.273 | | 0.391 | | **0.090** | | 0.500 | | 0.581 |
| Yes | 81 (15.0) | 48 (59.3) | | 59 (72.8) | | 37 (45.7) | | 24 (29.6) | | 6 (7.4) | |
| No | 459 (85.0) | 252 (54.9) | | 344 (74.9) | | 250 (54.5) | | 133 (29.0) | | 35 (7.6) | |
| | | | | | | 3c- Arab Women (N = 327) | | | | | |
| **Any IPV** | | | 0.516 | | **0.001** | | 0.547 | | **0.027** | | 0.073 |
| Yes | 218 (66.7) | 117 (53.7) | | 174 (79.8) | | 88 (40.4) | | 87 (39.9) | | 34 (15.6) | |
| No | 109 (33.3) | 59 (54.1) | | 69 (63.3) | | 44 (40.4) | | 31 (28.4) | | 10 (9.2) | |
| **Physical and/or sexual IPV** | | | 0.460 | | 0.326 | | 0.530 | | 0.556 | | 0.466 |
| Yes | 33 (10.1) | 17 (51.5) | | 23 (69.7) | | 13 (39.4) | | 12 (36.4) | | 5 (15.2) | |
| No | 294 (89.9) | 159 (54.1) | | 220 (74.8) | | 119 (40.5) | | 106 (36.1) | | 39 (13.3) | |
| **Emotional and/or verbal IPV** | | | 0.187 | | **0.004** | | 0.071 | | **0.025** | | **0.017** |
| Yes | 163 (50.0) | 83 (50.9) | | 132 (81.0) | | 59 (36.2) | | 68 (41.7) | | 29 (17.8) | |
| No | 163 (50.0) | 92 (56.4) | | 110 (67.5) | | 73 (44.8) | | 50 (30.7) | | 15 (9.2) | |
| **Social and economic IPV** | | | 0.304 | | **0.001** | | 0.490 | | 0.413 | | 0.142 |
| Yes | 165 (50.5) | 86 (52.1) | | 135 (81.8) | | 66 (40.0) | | 61 (37.0) | | 26 (15.8) | |
| No | 162 (49.5) | 90 (55.6) | | 108 (66.7) | | 66 (40.7) | | 57 (35.2) | | 18 (11.1) | |

an ER was not different among women who experienced any IPV and types of IPV, compared to those who experienced no IPV.

Different patterns of the associations between IPV and HCS use were observed in the Jewish and Arab study groups (Table 3B and 3C). Generally, among Jewish women, HCS utilization was lower among those who reported experiencing IPV compared to women not experiencing IPV. Visiting a family physician was more likely among Jewish women who

reported not experiencing physical and/or sexual IPV compared to women experiencing physical and/or sexual IPV (56.5% vs. 21.4%, respectively). Similar trends were found among Jewish women not experiencing IPV compared to Jewish women experiencing other forms of IPV, when we looked at the association between social and economic IPV and use of specialist services (54.5% vs. 45.7%, respectively) and between any IPV and physical and/or sexual IPV, and visiting an ER (31% vs. 23.4% respectively and 29.7% vs. 7.1% respectively)(Table 3B).

Among Arab women, the trend of the directions of the associations was different. Arab women who reported any IPV consistently made more visits to a gynecologist and the ER and were more likely to be hospitalized. Visiting a gynecologist was more frequent among Arab women who reported any IPV (79.8%), emotional and/or verbal IPV (81%), and social and economic IPV (81.8%), compared to Arab women who did not report these types of IPV (63.3%, 67.5% and 66.7%, respectively, for these IPV types). Visiting the ER was more likely among Arab women who reported any IPV and emotional and/or verbal IPV (39.9% and 41.7%, respectively) compared to women who reported not experiencing these types of IPV (28.4% and 30.7%, respectively). Hospitalization followed a similar trend, where Arab women who reported experiencing any IPV and emotional and/or verbal IPV were more likely to be hospitalized (15.6% and 17.8%, respectively) compared to women not experiencing IPV (9.2% and 9.2%, respectively). However, visiting a specialist was the exception: Arab women who did not report emotional and/or verbal IPV visited a specialist more often compared to women who reported this type of IPV (44.8% vs. 36.2%, respectively) (Table 3C).

## Associations between IPV screening, receiving information, and both, and HCS utilization

Table 4 presents the associations between ever been screened for IPV, received information about support services, and both, with HCS utilization for the total sample and for each study group. Only a few significant associations (P<0.01) were found. Ever been screened for IPV was associated with higher use of specialist services in the total sample and among Arab and Jewish women. However, in the total sample, ever been screened was associated with lower use of the ER. Also, in the total sample and among Arab women, those who reported both ever being screened and receiving information visited the ER less and were less likely to be hospitalized.

No significant associations were found between ever been screened for IPV, received information about support services, and both, with visiting a family doctor or a gynecologist.

A summary of the significance and direction of the univariate associations between any IPV, types of IPV, ever been screened for IPV, received information, and both, and use of HCS in the total sample and each study group can be found in S2 Appendix.

## Associations between women's characteristics and utilization of HCS

The univariate associations between women's characteristics and HCS utilization variables are presented in S1 Appendix. In the total sample, some characteristics were significantly associated with higher use of HCS (P<0.01). Use of family physician services was higher among unmarried women and those with chronic disease. Visiting a gynecologist was higher among pregnant women and women with no children. Utilization of specialist services was higher among women aged 25–34, women with university education, women who were after birth during the interview, and mothers of children. ER visit rates were higher among mothers in the youngest age group [16–24], with lower education (less than high school), those who were pregnant during the interview, and those with no children. Hospitalization was higher among women with lower education (high school or less), pregnant women, and mothers with no children during the interview (S1a Appendix).

**Table 4. Univariate Associations between IPV screening, receiving information, and both, and utilization of healthcare services in the total sample and study groups.**

| Women's Characteristics | Total | Family physician | | Gynecologist | | Specialist | | ER | | Hospitalization | |
|---|---|---|---|---|---|---|---|---|---|---|---|
| | N (%) | High use | P | Yes | P | Yes | P | Yes | P | Yes | P |
| | | N (%) | | N (%) | | N (%) | | N (%) | | N (%) | |
| **4a- Total sample (N = 869)** | | | | | | | | | | | |
| **IPV screening** | | | | | | | | | | | |
| Yes | 417 (48.3) | 228 (54.7) | 0.459 | 308 (73.9) | 0.358 | 224 (53.7) | **0.002** | 121 (29.0) | **0.058** | 35 (8.4) | 0.103 |
| No | 447 (51.7) | 247 (55.3) | | 336 (75.2) | | 194 (43.4) | | 153 (34.2) | | 50 (11.2) | |
| **Information received** | | | | | | | | | | | |
| Yes | 410 (47.2) | 224 (45.4) | 0.481 | 304 (74.1) | 0.460 | 203 (49.5) | 0.310 | 127 (31.0) | 0.389 | 34 (8.3) | **0.098** |
| No | 458 (52.8) | 206 (45.0) | | 342 (74.7) | | 218 (47.6) | | 147 (32.1) | | 51 (11.1) | |
| **Both screened and received information** | | | | | | | | | | | |
| Yes | 258 (29.9) | 136 (52.7) | 0.218 | 193 (74.8) | | 129 (50.0) | 0.299 | 67 (26.0) | **0.011** | 18 (7.0) | **0.039** |
| No | 605 (70.1) | 338 (55.9) | | 450 (74.4) | | 289 (47.8) | | 206 (34.0) | | 67 (11.1) | |
| **4b- Jewish women (N = 542)** | | | | | | | | | | | |
| **IPV screening** | | | | | | | | | | | |
| yes | 291 (54.0) | 161 (55.3) | 0.866 | 216 (74.2) | 0.837 | 167 (57.4) | **0.037** | 79 (27.1) | 0.273 | 21 (7.2) | 0.711 |
| no | 248 (46.0) | 139 (56.0) | | 186 (75.0) | | 120 (48.4) | | 78 (31.5) | | 20 (8.1) | |
| **Information received** | | | | | | | | | | | |
| yes | 291 (53.8) | 154 (52.9) | 0.225 | 219 (75.3) | 0.693 | 151 (51.9) | 0.489 | 86 (29.6) | 0.382 | 21 (7.2) | 0.427 |
| no | 250 (46.2) | 146 (58.4) | | 184 (73.6) | | 138 (55.2) | | 70 (28.0) | | 20 (8.0) | |
| **Both screened and received information** | | | | | | | | | | | |
| yes | 188 (34.9) | 102 (54.3) | 0.716 | 141 (75.0) | 0.917 | 103 (54.8) | 0.651 | 48 (25.5) | 0.232 | 13 (6.9) | 0.735 |
| no | 350 (65.1) | 197 (56.3) | | 260 (74.3) | | 184 (52.6) | | 108 (30.9) | | 28 (8.0) | |
| **4c- Arab women (N = 327)** | | | | | | | | | | | |
| **IPV screening** | | | | | | | | | | | |
| yes | 126 (38.8) | 67 (53.2) | 0.468 | 92 (73.0) | 0.364 | 57 (45.2) | **0.093** | 42 (33.3) | 0.249 | 14 (11.1) | 0.198 |
| no | 199 (61.2) | 108 (54.3) | | 150 (75.4) | | 74 (37.2) | | 75 (37.7) | | 30 (15.1) | |
| **Information received** | | | | | | | | | | | |
| yes | 119 (36.4) | 67 (53.2) | 0.104 | 85 (71.4) | 0.220 | 52 (43.7) | 0.208 | 41 (34.5) | 0.366 | 13 (10.9) | 0.200 |
| no | 208 (63.6) | 108 (54.3) | | 158 (76.0) | | 80 (38.5) | | 77 (37.0) | | 31 (14.9) | |
| **Both screened and received information** | | | | | | | | | | | |
| Yes | 70 (21.5) | 34 (48.6) | 0.194 | 52 (74.3) | 0.541 | 26 (37.1) | 0.32 | 19 (27.1) | **0.053** | 5 (7.1) | **0.052** |
| No | 255 (78.5) | 141(55.3) | | 190 (74.5) | | 105 (41.2) | | 98 (38.4) | | 39 (15.3) | |

Among Jewish women, visits to a family doctor were more likely among women who were after birth during the interview and women with chronic disease. Visiting a gynecologist was more frequent among women who were either pregnant or had no children during the interview. Visiting a specialist was likely among Jewish women with higher education, as well as among women who were after birth and who had children at the time of the interview. ER visits and hospitalization rates were higher among Jewish women who were pregnant and with no children during the interview (S1b Appendix).

For Arab women, high use of family physicians was more likely among women who were not mothers at the time of the interview. Use of a gynecologist was higher among Arab women who were pregnant and who had no children at the time of the interview. Visits to a specialist were more likely among women who were after birth at the time of the interview. Visiting the ER was likely among women who were pregnant, and women who had no children at the time of the interview. Hospitalization rate was higher among Arab women with postsecondary or

**Table 5. Multivariable analysis for associations between any IPV, types of IPV, and HCS utilization among Jewish and Arab women (adjusted models)[*].**

5a- Jewish women (N = 542)

| | Family physician | Gynecologist | Specialist | Hospitalization | ER |
|---|---|---|---|---|---|
| | AOR (95%CI) | AOR (95%CI) | AOR (95%CI) | AOR (95%CI) | AOR (95%CI) |
| **Any IPV** | | | | | |
| Yes | 0.90 (0.51–1.57) | 0.97 (0.62–1.53) | 0.89 (0.56–1.40) | 1.13 (0.55–2.36) | **0.62 (0.41–0.93)** |
| No | 1 | 1 | 1 | | 1 |
| **Physical and/or sexual IPV** | | | | | |
| Yes | **0.21 (0.05–0.82)** | 2.50 (0.36–17.38) | 0.72 (0.15–3.45) | NR | 0.18 (0.02–1.52) |
| No | 1 | 1 | 1 | 1 | 1 |
| **Social and economic IPV** | | | | | |
| Yes | 1.09 (0.68–1.75) | 0.84 (0.45–1.57) | 0.74 (0.43–1.25) | 0.76 (0.38–1.52) | 0.86 (0.52–1.40) |
| No | 1 | 1 | 1 | 1 | 1 |
| **Emotional and/or verbal IPV** | | | | | |
| Yes | 0.77 (0.39–1.54) | 1.18 (0.70–2.00) | 1.06 (0.64–1.76) | 1.28 (0.58–2.82) | 0.73 (0.46–1.16) |
| No | 1 | 1 | 1 | 1 | 1 |

5b- Arab women (N = 322)

| | family physician | Gynecologist | Specialist | Hospitalization | ER |
|---|---|---|---|---|---|
| | AOR (95%CI) | AOR (95%CI) | AOR (95%CI) | AOR (95%CI) | AOR (95%CI) |
| **Any IPV** | | | | | |
| Yes | 1.05 (0.65–1.69) | **2.00 (1.14–3.51)** | 1.10 (0.70–1.75) | 1.23 (0.34–4.41) | 1.36 (0.86–2.16) |
| No | 1 | 1 | 1 | 1 | |
| **Physical and/or sexual IPV** | | | | | |
| Yes | 1.26 (0.59–2.69) | 0.81 (0.35–1.91) | 1.10 (0.50–2.41) | NR | 1.07 (0.57–2.03) |
| No | 1 | 1 | 1 | | 1 |
| **Social and economic IPV** | | | | | |
| Yes | 0.92 (0.63–1.36) | **2.17 (1.23–3.81)** | 1.18 (0.65–2.15) | 1.22 (0.49–3.03) | 0.91 (0.53–1.54) |
| No | 1 | 1 | 1 | 1 | |
| **Emotional and/or verbal IPV** | | | | | |
| Yes | 0.94 (0.61–1.45) | **1.83 (1.04–3.22)** | 0.76 (0.50–1.16) | 1.95 (0.63–6.06) | 1.50 (0.96–2.36) |
| No | 1 | 1 | 1 | 1 | |

[*]All associations are adjusted for socio-demographic variables (age, education, chronic disease, and pregnancy status). Significant associations are highlighted with bold fonts.

college education, and among women who were pregnant at the time of the interview (S1c Appendix).

Next, and before we conducted the multivariable analysis for use of HCS variables, we examined correlations between the socio-demographic variables. No correlation was found higher than our threshold of 0.5 for the correlation coefficient. Therefore, all of these variables were included in the multivariable analysis, as multi-collinearity was not likely (S3 Appendix).

**Multivariable associations for HCS utilization and IPV.** Table 5 presents results of the multivariable analysis for HCS utilization among Jewish and Arab women. The odds ratios (ORs) and 95% confidence intervals (CI) present adjusted associations between an IPV variable (any IPV, physical and/or sexual IPV, emotional and/or verbal IPV, and social and economic IPV) and one HCS utilization variable (visits to family physician, gynecologist, specialist, ER, and hospitalization). All associations were adjusted for age, education, chronic disease, and pregnancy status at the time of interview. After these adjustments, few associations between any IPV, as well as specific IPV types and HCS utilization remained significant.

Also, we found different patterns of HCS utilization among women experiencing IPV compared to those not experiencing IPV in each ethnic group (Table 5).

Among Jewish women (Table 5A), experiencing physical and/or sexual IPV compared to not experiencing this type of IPV was associated with lower use of family physician services (OR, 95%CI = 0.21, 0.05–0.82). A similar trend was found for ER visits: compared to Jewish women who reported not experiencing any IPV, Jewish women who reported experiencing any IPV were less likely to visit the ER (AOR, 95%CI = 0.62, 0.41–0.93). The rest of the associations were not significant in the multivariate associations of HCS utilization among Jewish women.

Among Arab women, a different trend was observed (Table 5B). Arab women who reported any IPV, emotional and/or verbal IPV, and social and economic IPV were more likely to use gynecologist services (AOR, 95%CI = 2.00, 1.14–3.51; 2.17, 1.23–3.81, and 1.83, 1.04–3.22) compared to Arab women who reported not experiencing these types of IPV. The other associations between any IPV, types of IPV, and the HCS utilization variables were no longer significant among Arab women.

### The effect of IPV screening, receiving information, and both, on the associations between IPV and HCS utilization

Here we were interested in the effect of IPV screening, receiving information on supportive services, and both, on the associations between any IPV, types of IPV, and HCS utilization. We therefore conducted multivariable analysis for the associations that were found to be significant between these variables in each group of women separately. Table 6 presents two sets of multivariable models for Jewish women. The first shows associations between physical and/ or sexual IPV and visits to a family physician (Table 6A), and the second shows associations between any IPV and ER use (Table 6B). For both tables the associations between IPV and HCS utilization remained significant and in the same direction in all models: Jewish women who reported IPV used less HCS. Adjustment for IPV screening, receiving information, or both, did not change the associations between IPV and HCS utilization. The variables of screening and receiving information were not significant in the models in eitherTable 6A and 6B.

Table 7 presents the associations between any IPV (Table 7A), emotional and/or verbal IPV (Table 7B), and social and economic IPV (Table 7C) and use of gynecologist services among Arab women. All of these associations were positive, where women who reported experiencing any of these IPV variables were more likely to use a gynecologist's services compared to women who reported not experiencing any of these.

Introducing IPV screening and receiving information into Models 3 and 4 did not change the strength of the associations compared to Model 2. However, in Model 5, the index variable of both having been screened for IPV and receiving information was significant, though it contributed little to decreasing the strength of the associations (OR) compared to Model 2.

## Discussion

To our knowledge, this is the first study to compare patterns of HCS utilizations among women experiencing IPV versus non-victims, and to examine the effects of IPV screening and receiving information on these patterns among women from different ethnic groups.

### HCS utilization patterns in relation to IPV differ by ethnicity

A main finding of the current study relates to the complexity of the association between IPV and HCS utilization. While previous research suggested higher HCS utilization among women

**Table 6. Multivariate associations between experiencing physical and/or sexual IPV and health care utilization among Jewish women (N = 542).**

**6a- Multivariable associations between *physical and/or sexual IPV* and *visits to family physicians* among Jewish women (N = 542)**

| | Model 1 | Model 2 | Model 3 | Model 4 | Model 5 |
|---|---|---|---|---|---|
| | OR (95%CI) | OR (95%CI) | OR (95%CI) | OR (95%CI) | OR (95%CI) |
| **Physical and/or sexual IPV** | | | | | |
| Yes | **0.20 (0.05–0.83)** | **0.19 (0.04–0.80)** | **0.20 (0.05–0.80)** | **0.19 (0.04–0.75)** | **0.19 (0.49–0.75)** |
| No | 1 | 1 | 1 | 1 | 1 |
| **Screened for IPV** | | | | | |
| Yes | | | 1.08 (0.82–1.41) | | |
| No | | | 1 | | |
| **Received information** | | | | | |
| Yes | | | | 1.32 (0.90–1.93) | |
| No | | | | 1 | |
| **Both screened and received information** | | | | | |
| Yes | | | | | 1.19 (0.84–1.67) |
| No | | | | | 1 |

**6b- Multivariable associations between *any IPV* and *ER visits* among Jewish women (N = 542)**

| | Model 1 | Model 2 | Model 3 | Model 4 | Model 5 |
|---|---|---|---|---|---|
| | OR (95%CI) | OR (95%CI) | OR (95%CI) | OR (95%CI) | OR (95%CI) |
| **Any IPV** | | | | | |
| Yes | **0.66 (0.45–0.96)** | **0.62 (0.41–0.93)** | **0.62 (0.41–0.94)** | **0.63 (0.42–0.95)** | **0.61 (0.40–0.93)** |
| No | 1 | 1 | 1 | 1 | 1 |
| **Screened for IPV** | | | | | |
| Yes | | | 1.14 (0.76–1.72) | | |
| No | | | 1 | | |
| **Received information** | | | | | |
| Yes | | | | 0.81 (0.58–1.23) | |
| No | | | | 1 | |
| **Both screened and received information** | | | | | |
| Yes | | | | | 1.23 (0.74–2.05) |
| No | | | | | 1 |

Model 1- crude model. Model 2- adjusted for socio-demographic variables (age, education, chronic illness and pregnancy status). Model 3- adjusted for socio-demographic variables and IPV screening. Model 4- adjusted for socio-demographic variables and received information. Model 5- adjusted for socio-demographic variables and a combined variable of both being screened for IPV and received information.

experiencing IPV compared to non-victims [13, 18], we found different patterns of HCS utilization in relation to IPV among women from the study's two ethnic groups (Arab and Jewish), with diverse directions for the associations between IPV and five HCS utilization variables.

We observed positive associations between IPV variables and HCS utilization among Arab women, while these associations were negative among Jewish women. While these various patterns might attest to different patterns of HCS utilization in the general Arab and Jewish populations in Israel [54, 56, 58], it might also indicate dissimilar help seeking patterns among Arab women experiencing IPV [61, 62]. It could be that high use of medical HCS among Arab women in our study points to access problems in seeking help through social and mental HCS. Arab women might be reluctant to use these HCS, as such services are often stigmatized in Israel's Arab community [67] and because using them would imply IPV disclosure [61, 68]. Given a lack of proper societal solutions and support services for all women in Israel and specifically Arab women experiencing IPV, disclosure of IPV to HCS or social services providers

**Table 7. Multivariate associations between experiencing IPV and HCS utilization among Arab women (N = 322).**

**7a- Multivariable associations between *any IPV* and use of *gynecologist services* among Arab women (N = 322)**

| | Model 1 | Model 2 | Model 3 | Model 4 | Model 5 |
|---|---|---|---|---|---|
| | OR (95%CI) | OR (95%CI) | OR (95%CI) | OR (95%CI) | OR (95%CI) |
| **Any IPV** | | | | | |
| Yes | **2.29 (1.42–3.71)** | **1.93 (1.14–3.29)** | **1.96 (1.15–3.34)** | **1.93 (1.13–3.30)** | **1.90 (1.09–3.30)** |
| No | **1** | **1** | **1** | **1** | **1** |
| **Screened for IPV** | | | | | |
| Yes | | | 0.64 (0.38–1.08) | | |
| No | | | 1 | | |
| **Received information** | | | | | |
| Yes | | | | 0.93 (0.60–1.40) | |
| No | | | | 1 | |
| **Both screened and received information** | | | | | |
| Yes | | | | | 0.61 (0.39–0.97) |
| No | | | | | 1 |

**7b - Multivariable associations between *social and economic IPV* and use of *gynecologist services* among Arab women (N = 322)**

| | Model 1 | Model 2 | Model 3 | Model 4 | Model 5 |
|---|---|---|---|---|---|
| | OR (95%CI) | OR (95%CI) | OR (95%CI) | OR (95%CI) | OR (95%CI) |
| **Social and economic IPV** | | | | | |
| Yes | **2.16 (1.32–3.52)** | **1.89 (1.18–3.03)** | **1.91 (1.17–3.13)** | **1.88 (1.17–3.02)** | **1.83 (1.14–2.95)** |
| No | **1** | **1** | **1** | **1** | **1** |
| **Screened for IPV** | | | | | |
| Yes | | | 0.64 (0.38–1.09) | | |
| No | | | 1 | | |
| **Received information** | | | | | |
| Yes | | | | 0.91 (0.60–1.39) | |
| No | | | | 1 | |
| **Both screened and received information** | | | | | |
| Yes | | | | | 0.62 (0.39–0.98) |
| No | | | | | 1 |

**7c - Multivariable associations between *emotional and/or verbal IPV* and use of *gynecologist services* among Arab women (N = 322)**

| | Model 1 | Model 2 | Model 3 | Model 4 | Model 5 |
|---|---|---|---|---|---|
| | OR (95%CI) | OR (95%CI) | OR (95%CI) | OR (95%CI) | OR (95%CI) |
| **Emotional and/or verbal IPV** | | | | | |
| Yes | **2.07 (1.27–3.98)** | **1.94 (1.12–3.33)** | **1.97 (1.51–3.37)** | **1.94 (1.13–3.33)** | **1.94 (1.13–3.34)** |
| No | **1** | **1** | **1** | **1** | **1** |
| **Screened for IPV** | | | | | |
| Yes | | | 0.63 (0.38–1.06) | | |
| No | | | 1 | | |
| **Received information** | | | | | |
| Yes | | | | 0.92 (0.60–1.40) | |
| No | | | | 1 | |
| **Both screened and received information** | | | | | |
| Yes | | | | | 0.58 (0.36–0.95) |
| No | | | | | 1 |

might be very complex for an Arab woman, as it could lead to elevated problems not only in her marriage, but also in her and her husband's extended families [61]. Disclosure could also endanger these women's wellbeing and that of their children [61], as safety planning is not on

offer and homicide is on the increase among Arab women in recent decades [47]. High use of medical HCS among Arab women experiencing IPV might also indicate lack of access to social and mental HCS. Mental HCS are lacking in Israel's Arab population in general. There are few mental health clinics, and even fewer psychologists and psychiatrists [57, 58] who can provide help to these women in their own language (Arabic). Therefore, Arab women might seek help through medical HCS multiple times as a sign of distress, the IPV being ongoing, but their non-disclosure of IPV resulting in a failure to be referred on for IPV-specific care.

As for Jewish women, they might seek less help through medical HCS because they find more solutions related to their IPV through social and mental HCS, given that they face fewer barriers in seeking help at these HCS compared to Arab women [58, 67]. However, Jewish women experiencing IPV might still face barriers in seeking help through medical HCS, as perpetrators might prevent them from doing so, as was shown among women experiencing IPV in the US [25].

## Utilization patterns associated with IPV for specific HCS

We found various patterns of HCS utilization among Arab and Jewish women for the specific HCS. Future investigations are needed to explore the factors behind these patterns, but they might relate to several factors, such as women's perceptions about a given service [25], and level of trust in providers within different HCS [18]. While women experiencing IPV might seek help via HCS based on trust in healthcare providers [23], they might nonetheless be reluctant to disclose IPV to these providers out of shame, or fear that the provider might know the woman's partner/perpetrator [25]. Another factor possibly leading to different utilization of the HCS is the type of IPV women have been experiencing, as well as length of exposure. For example, Benomi et al. [13], found that HCS utilization was higher among women who experience physical and/or sexual compared to non-physical or sexual abuse, and among women with ongoing abuse compared those with recent abuse [13].

Regarding family physician services, we found lower utilization among Jewish women who reported physical and/or sexual IPV compared to women who did not report experiencing this physical and/or sexual IPV, and no significant association with IPV among Arab women. Lower visits to a family physician by Jewish women might have different explanations. Some research showed that while most women trust their family physician, women experiencing IPV might feel shame or be afraid to disclose IPV and, as such, make fewer visits to them as their fear overrides their trust [16]. As well, a woman who discloses IPV to her family doctor may be referred on to proper counseling or mental or social services and is less likely to conduct repeated visits to this doctor for IPV. This might explain the negative association we found between physical and/or sexual IPV and visits to a family physician (but not for social and economic or emotional and/or verbal IPV) among Jewish women, as women might find it easier to have this type of IPV diagnosed and to get such a referral.

As for Arab women experiencing IPV, they tend to use family physicians less. This might be due to cultural barriers. For many Arab women, the family physician in their community is a family relative [60] who might know her and her abusive partner. Arab women might also feel ashamed disclosing IPV to their family physician and might trust them less on this issue than on others. Thus, if an Arab woman is not interested in disclosing IPV, she is unlikely to visit her family physician.

Regarding the use of a gynecologist's services, our results among Arab women revealing high use are consistent with previous findings showing that women experiencing IPV face more problems related to reproductive health [2, 69]. As such, they are more likely to use gynecological services. Arab women might also trust their gynecologist more (than a family

physician) because healthcare management organizations (HMOs) try to appoint more female gynecologists at clinics in Arab towns and villages, which might reduce cultural barriers for utilization of this service and create a refuge for Arab women. Since gynecological services are not specific to IPV, women might also return to their gynecologist while IPV is ongoing without being referred on to other proper services. Lower use of gynecologist services among Jewish women might follow the above explanation: if Jewish women are referred on to social or mental health support services, they are less likely to use the medical services, including a gynecologist. Future research is needed to explore this possibility, as the current analysis focused on medical HCS.

Lower use of specialist care in both Jewish and Arab women who reported experiencing IPV might indicate that this service provides fewer solutions to IPV related problems, particularly since the data on specialist use concerned surgeons and internists, which are not typical services in cases of IPV, therefore Arab and Jewish women might visit them less.

As for ER visits and hospitalization, women experiencing IPV are more likely to be hospitalized, as they might suffer severe injuries that require time to recover from [2]. High use of the ER and hospitalizations among Arab women experiencing IPV might speak to the severity of IPV problems in this group, leading to more hospitalization. The fact that Jewish women experiencing any IPV in our study used the ER less often compared to Jewish women who experience no IPV might be explained by their use of other community HCS, which reduces their need for ER visits. Alternatively, IPV among them might be less severe compared to Arab women. These hypotheses should be further examined using both quantitative and qualitative methodologies.

## The effect of IPV screening and information received on support services and HCS utilization

Generally, this study revealed no effect of IPV screening and receiving information on the association between IPV and HCS utilization for the total sample, nor for Jewish women. However, for Arab women we found a small, significant effect of the index variable (i.e., both screened for IPV and received information) on the associations between IPV types and use of gynecologist services. This might indicate that if a gynecologist conducts IPV screening, provides women with information on support services, then refers them on to proper social or mental health services, women experiencing IPV will likely use the gynecologist services less. This highlights the importance of conducting both IPV screening and providing information for Arab women through HCS. However, in this study as well as in our previous study [52], we found lower reports among Arab women of IPV screening and receiving information through HCS compared to Jewish women. Previous research showed that IPV screening was important for women who visit the ER [70]. While there is a debate about universal screening for IPV and universal screening for safety and providing universal information and education on IPV [71, 72], it is agreed conducting any form of screening can help women disclose abuse, and help healthcare providers to properly respond to, and treat not only the physical and/or sexual, but also the psychological aspects of IPV [26, 73, 74]. Among Jewish women, no association was found between IPV screening and receiving information. This suggests that these interventions should be more consistent among these women. Improving screening and information provision among women experiencing IPV in Israel is vital, particularly given that both have been mandatory in the country since 2003 [50], but while gaps in implementation, especially among minority women, have already been identified [52]. Gaps might relate to lack of clarity about the country's screening strategy and which tools should be used, as well as staff training issues and other barriers that have previously been identified in research among HCP

in Israel among [51]. These barriers might be removed by system-level interventions, as was suggested in a recent systematic review [46]. However, more research is needed in order to learn about tailored strategies and interventions appropriate for HCS in Israel to help women experiencing IPV, particularly as debate remains over the effectiveness of universal screening policies [37].

**Study limitations.** While this is the first study we know of to examine patterns of medical HCS utilization among women experiencing and not experiencing IPV, and to explore the effects of IPV screening and receiving information on these patterns, the research has some limitations. First, only data on medical HCS utilization were included, and not on use of social, psychological, or mental health services. Information on these services was not provided by Clalit Health Services. That's because a period of reform in the area of mental health in Israel had just begun in 2015, which led to mental HCS being provided in the community by health funds [58]. Before this, mental HCS services were provided by clinics run by the MOH. Information on mental and social HCS services would have provided a more comprehensive picture of HCS utilization among women with IPV. Future research should aim to provide this fuller picture of the use of mental and social HCS in addition to medical HCS in relation to IPV.

Second, our IPV measures were self-reported, which might have underestimated this phenomenon. Further, IPV measures were not time specific, as we asked about IPV and any IPV types in general, and that was before follow-up regarding HCS utilization. However, studies show that IPV is often a lifelong problem [75–78]. While physical and/or sexual IPV decreases during pregnancy, emotional and/or verbal, and social and economic IPV continue, and, many times, physical and/or sexual IPV begins again after birth [79]. Since our participants were pregnant or 6 weeks to 6 months after birth, it is possible that IPV continued after birth. Third, our information on screening of IPV and receiving information was similarly not time limited. The questions in our original study asked about ever receiving these services. As such, reports on these variables might be prone to recall bias.

**Conclusions and implications for practice.** We found complex patterns of HCS utilization among women experiencing IPV that varied by ethnicity. Differences in this regard between Arab and Jewish women in Israel might be due to the differential ability of these services to address the multiple health needs of women who experience IPV in a community versus hospital setting. The small positive effect of having been both screened for IPV and having received information on supportive services on the association between IPV and HCS utilization among Arab women points to an opportunity to help women experiencing IPV. Health policy and planning to address the needs of women experiencing IPV should consider the complexity of the patterns of HCS utilization among women experiencing IPV and provide them with tailored solutions.

## Supporting information

**S1 Appendix. Associations between women's characteristics and healthcare utilization measures at first-year follow-up in the total sample and the study groups (Arab and Jewish women).**
(DOCX)

**S2 Appendix. Summary of univariate associations[b] between IPV variables, IPV screening, information received, and both, and HCS utilization variables[a].**
(DOCX)

**S3 Appendix. Correlations between study variables.**
(DOCX)

## Acknowledgments

We thank the women who agreed to participate in the study.

## Author Contributions

**Conceptualization:** Nihaya Daoud, Ilana Shoham-Vardi, Nadav Davidovitch, Arnon Cohen.

**Data curation:** Nihaya Daoud, Ruslan Sergienko, Naama Batat, Ilana Shoham-Vardi, Arnon Cohen.

**Formal analysis:** Nihaya Daoud, Lotan Kraun, Ruslan Sergienko, Naama Batat.

**Funding acquisition:** Nihaya Daoud.

**Investigation:** Nihaya Daoud, Lotan Kraun, Ruslan Sergienko, Naama Batat, Ilana Shoham-Vardi.

**Methodology:** Nihaya Daoud, Lotan Kraun.

**Project administration:** Nihaya Daoud.

**Resources:** Nihaya Daoud, Arnon Cohen.

**Software:** Nihaya Daoud.

**Supervision:** Nihaya Daoud.

**Validation:** Nihaya Daoud.

**Visualization:** Nihaya Daoud, Nadav Davidovitch.

**Writing – original draft:** Nihaya Daoud.

**Writing – review & editing:** Nihaya Daoud, Lotan Kraun, Ruslan Sergienko, Naama Batat, Ilana Shoham-Vardi, Nadav Davidovitch, Arnon Cohen.

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
