## [Decision Letter · Decision Letter 0]

11 Sep 2019

PONE-D-19-17815

Patterns of healthcare services utilization associated with intimate partner violence (IPV): effects of IPV screening and receiving information among a cohort of women

PLOS ONE

Dear Dr. Daoud,

Thank you for submitting your manuscript to PLOS ONE. After careful consideration, we feel that it has merit but does not fully meet PLOS ONE’s publication criteria as it currently stands. Therefore, we invite you to submit a revised version of the manuscript that addresses the minor points raised by both reviewers during the review process.

We would appreciate receiving your revised manuscript by Oct 26 2019 11:59PM. To enhance the reproducibility of your results, we recommend that if applicable you deposit your laboratory protocols in protocols.io, where a protocol can be assigned its own identifier (DOI) such that it can be cited independently in the future. For instructions see: http://journals.plos.org/plosone/s/submission-guidelines#loc-laboratory-protocols

We look forward to receiving your revised manuscript.

Kind regards,

Soraya Seedat

Academic Editor

PLOS ONE

Journal Requirements:

Reviewers' comments:

Reviewer's Responses to Questions

**Comments to the Author**

1. Is the manuscript technically sound, and do the data support the conclusions?

Reviewer #1: Yes

Reviewer #2: Yes

2. Has the statistical analysis been performed appropriately and rigorously? 

Reviewer #1: Yes

Reviewer #2: Yes

3. Have the authors made all data underlying the findings in their manuscript fully available?

Reviewer #1: Yes

Reviewer #2: Yes

4. Is the manuscript presented in an intelligible fashion and written in standard English?

Reviewer #1: Yes

Reviewer #2: Yes

5. Review Comments to the Author

Reviewer #1: Thank you for submitting this very important manuscript about intimate partner violence and healthcare services utilization. Overall, this is a very well-written and well-developed manuscript. I appreciate the intention of specifically teasing apart the differences in healthcare service utilization among two cultural groups living in the same country. I have a few suggestions throughout the manuscript that could use some additional details and clarification to help the reader fully interpret the results.

Throughout Manuscript:

1) Since this manuscript focuses exclusively on pregnant and postpartum women, I would amend the title to note that, “Patterns of healthcare service utilization associated with intimate partner violence (IPV): Effects of IPV screening and receiving information among a cohort of perinatal women.

2) Throughout the manuscript I would stress that this is “self-identified” IPV and that you still likely have an underestimation of the true prevalence among study participants.

3) The term women/woman implies gender; it appears your subjects were are of the female sex (as they were able to conceive). I think it would be important to consider your use of gender/sex.

Introduction:

4) The term social IPV is introduced on page 5, but a definition of “social IPV” is not provided until page 10. Furthermore, there are numerous points throughout the manuscript where social/economic IPV are combined and then others where they are not. Can you please clarify the use of social/economic IPV earlier in the manuscript?

5) I would also encourage the authors to take a look at the recent review on provided screening and counseling for IPV (Alvarez, Fedock, Grace, & Campbell, 2017, Trauma, Violence, & Abuse) for the introduction section.

Methods:

6) How was the cut-off for healthcare services utilization determined (e.g., low versus high use)? Since many of the participants were enrolled during pregnancy, then wouldn’t they automatically be “high-use” patients related to their antepartum care visits? How was this factored in?

7) How did you determine your age groups? Why was age 48 included if your highest age was 44?

8) Why was the level of significance set at p=.10? It appears you were powered to use a more traditional p-value of .05.

Discussion

9) There are currently debates about the usefulness of universal screening for IPV versus universal screening for safety or providing universal education (e.g., Jack, Ford-Gilboe, Davidov, & MacMillan, 2017, Journal of Clinical Nursing; Miller, Breach, & Thurston, 2018, JAMA Internal Medicine). How do these debates factor into your findings and the current policies in Israel?

10) Prevalence of abuse during pregnancy varies (Devries et al., 2010, Reproductive Health Matters) and can be dependent on age, gravida, and location. I think it would also be important to cite literature that looks at the global picture beyond Canada.

Reviewer #2: I want tot commend you on the submission of your study titled: Patters of healthcare service utilization with intimate partner violence: effects of IPV screening and receiving information among a cohort of women.

The author(s) present a robust manuscript worth of publication.

I have identified several areas that require clarification.

Title: clarify type of information being provided or received. The term information is too vague.

List of abbreviations: add VAW Violence Against Women

Introduction: substitute diseases in line 70 for infections.

Define intimate partner violence in the introduction section.

The term 'victim' is underscored in your study. What about the term survivor. Why was this term not used or is it not applicable? if not why not?

Line 86 missing 'of' between many these.

Line 91: elaborate on what getting help means.

Line 92: consider using the term trust or confidence instead of faith in the health care system.

Line 121: consider using the term women experiencing violence rather that victims.

Line 127: did you account for sexual violence and if not why not. Elaborate.

Line 136: social and personal barriers. Please elaborate.

Line 181: use (n=1401)

Line 220: close spacing between study and ,

Study measures: well described. Salient and consistent with study intent.

Results: presented clearly. Adequate statistical tests and models selected.

table 2. Line 309. May want to draw line after Arab women to highlight sample.

Line 315-316: use %'s.

Again if sexual abuse was accounted for include results. Line 318-322

Multivariate associations drawn are accurate and consistent with research aims.

Line 488: table 7 Include the n= in table heading to keep consistent with preservation of other tables.

Discussion sections is well articulated as well.

Limitations are described and cited.

I would like to see the author(s) elaborate on the implications for practice and policy.

Again, I was to thank you for your valuable submission and encourage you to address the questions and comments.

Regards -

6. PLOS authors have the option to publish the peer review history of their article (what does this mean?). If published, this will include your full peer review and any attached files.

Reviewer #1: No

Reviewer #2: Yes: Eva Margarita Moya

---

## [Author Response · Author response to Decision Letter 0]

15 Oct 2019

PLOS ONE

Professor Soraya Seedat, 

Academic Editor

Dear Prof. Seedat, 

Thank you very much for your letter dated Sept. 26th informing us of the results of your review of our paper, “Patterns of healthcare services utilization associated with intimate partner violence (IPV): effects of IPV screening and receiving information among a cohort of women (PONE-D-19-17815). Following the reviewer’s comments the title is now: “Patterns of healthcare services utilization associated with intimate partner violence (IPV): effects of IPV screening and receiving information on support services in a cohort of perinatal women”.

We would also like to thank the reviewers for their valuable comments and suggestions. We have responded to these and provide point-by-point answers below. We used track changes to highlight changes to the original manuscript, and point to these in our response with page and line numbers. 

Dr. Nihaya Daoud 

Corresponding author 

Answer to reviewers’ comments:

Answers to Reviewer #1: 

Comment 1:

Thank you for submitting this very important manuscript about intimate partner violence and healthcare services utilization. Overall, this is a very well-written and well-developed manuscript. I appreciate the intention of specifically teasing apart the differences in healthcare service utilization among two cultural groups living in the same country. I have a few suggestions throughout the manuscript that could use some additional details and clarification to help the reader fully interpret the results.

Answer to comment 1:

We thank the reviewer very much for the positive evaluation of our paper and for the valuable comments and suggestions, which have improved our manuscript. 

Throughout Manuscript:

Comment 2 Since this manuscript focuses exclusively on pregnant and postpartum women, I would amend the title to note that, “Patterns of healthcare service utilization associated with intimate partner violence (IPV): Effects of IPV screening and receiving information among a cohort of perinatal women.

Answer to comment 2: Thank you for your suggestion. We have added ‘perinatal’ to the title.

Comment 3- Throughout the manuscript I would stress that this is “self-identified” IPV and that you still likely have an underestimation of the true prevalence among study participants.

Answer to comment 3: Our revised Abstract, study aims, and Methods section (p. 7 and p. 10) now clarify that our measure of IPV is self-reported. Further, in our Discussion, we have added the point that our results might underestimate true prevalence in the study sample (see p. 30). 

Comment 4. The term women/woman implies gender; it appears your subjects were are of the female sex (as they were able to conceive). I think it would be important to consider your use of gender/sex. 

Answers to comment 4: . This is an important comment. After consideration we decided to use the term ‘women’ across the paper, since this term is used in many biomedical papers in context of the perinatal period and we think that since all participants were either pregnant or postpartum it is clear that all participants were female. 

Comments on the Introduction:

Comment 5. The term social IPV is introduced on page 5, but a definition of “social IPV” is not provided until page 10. Furthermore, there are numerous points throughout the manuscript where social/economic IPV are combined and then others where they are not. Can you please clarify the use of social/economic IPV earlier in the manuscript?

Answer to comment 5: Thank you for your attention. We have now changed the term ‘social IPV’ to ‘social and economic IPV’ throughout the paper for clarity. 

Comment 6. I would also encourage the authors to take a look at the recent review on provided screening and counseling for IPV (Alvarez, Fedock, Grace, & Campbell, 2017, Trauma, Violence, & Abuse) for the introduction section.

Answer to comment 6: Thank for suggesting the review by Alvarez, Fedock, Grace, & Campbell, 2017, Trauma, Violence, & Abuse. We have now included this paper in our Introduction (lines 121-127, page 5).

Comments on Methods:

Comment 7. How was the cut-off for healthcare services utilization determined (e.g., low versus high use)? Since many of the participants were enrolled during pregnancy, then wouldn’t they automatically be “high-use” patients related to their antepartum care visits? How was this factored in?

Answer to comment 7: Our Measures section describes how we determined the cut-off for HCS utilization categories, as follows: 

“After examining the distribution of HCS use variables, we dichotomized data into yes (used the services) or no (did not use the service) by the median score (=0) for four of five variables. For the fifth variable, related to visits to family physician, we dichotomized data as follows: 1. low use (0-4 visits for a period of one year) as we estimated one visit per 3 months, and 2. high use (5 or more visits during a period of one year).” Please see p. 10, lines 257-261.

Comment 8. How did you determine your age groups? Why was age 48 included if your highest age was 44?

Answer to comment 8: Thank you for your attention. The age groups were determined by the distribution of women in the groups. For the original study, the highest age of the women was 48 (see p. 8). For the current sample, which only included women members of one large sick fund, the highest age was 44 years. We have now clarified this in our Measures section (lines 278-279, p. 11). 

Comment 9. Why was the level of significance set at p=.10? It appears you were powered to use a more traditional p-value of .05.

Answer to comment 9: We decided to use statistical significance of P=0.1 because some univariate associations were borderline, and we were interested in including associations in the multivariable analysis variables that might have introduced residual confounding. 

Comments on the Discussion

Comment 10. There are currently debates about the usefulness of universal screening for IPV versus universal screening for safety or providing universal education (e.g., Jack, Ford-Gilboe, Davidov, & MacMillan, 2017, Journal of Clinical Nursing; Miller, Breach, & Thurston, 2018, JAMA Internal Medicine). How do these debates factor into your findings and the current policies in Israel?

Answer to comment 10. 

Thanks for your comment. We have now added this point to the Discussion (p. 29, lines 628-632 for the debate and page 30, lines 634-644 for the gaps in screening in Israel). 

Comment 11. Prevalence of abuse during pregnancy varies (Devries et al., 2010, Reproductive Health Matters) and can be dependent on age, gravida, and location. I think it would also be important to cite literature that looks at the global picture beyond Canada.

Answer to comment 11. 

Thank you for this suggestion. We have included this citation now. 

Answer to Reviewer #2 comments: 

I want to commend you on the submission of your study titled: Patters of healthcare service utilization with intimate partner violence: effects of IPV screening and receiving information among a cohort of women.

Comment 1. The author(s) present a robust manuscript worth of publication.

Answer to comment 1. We thank the author for the positive evaluation and for indicating the robustness of our paper. 

I have identified several areas that require clarification.

Comment 1, Title: clarify type of information being provided or received. The term information is too vague.

Answer to comment 1: The title is already very long. However, we have added “information on support services.” 

Comment 2, List of abbreviations: add VAW Violence Against Women

Answer to comment 2: We have added this to the list that appears on p. 1, thank you. 

Comment 3, Introduction: substitute diseases in line 70 for infections.

Answer to comment 3: Thank you we have changed to STIs (line 74 now). 

Comment 4: Define intimate partner violence in the introduction section.

Answer to comment 4, We have added a definition of intimate partner violence (IPV) (p.3, lines 71-73).

Comment 5: The term 'victim' is underscored in your study. What about the term survivor. Why was this term not used or is it not applicable? if not why not?

Answer to comment 5: We have changed our wording to “women experiencing intimate partner violence.” We chose this term because many of the women we interviewed are still living with the perpetrator and might be in an ongoing, abusive relationship. Usually, the term “survivors” is used for women who leave an abusive relationship. 

Edits comment: 

Comment: Line 86 missing 'of' between many these.

Answer: Thank you for your attention. We have added this (now line 92). 

Comment: Line 91: elaborate on what getting help means.

Answer: We have changed this to “professional support services” (now line 97). 

Comment: Line 92: consider using the term trust or confidence instead of faith in the health care system.

Answer: Thank you. We have changed this accordingly (now line 99). 

Comment: Line 121: consider using the term women experiencing violence rather that victims.

Answer: We have changed this the term “victims of IPV” to “women experiencing IPV” across the paper. 

Comment: Line 127: did you account for sexual violence and if not why not. Elaborate.

Answer: In the current paper, sexual violence by an intimate partner was included in physical violence. We have now clarified this in our Measures section (i.e., that physical IPV includes sexual IPV). This choice results from a factor analysis we conducted in a previous analysis (see: Daoud N, Sergienko R, Shoham-Vardi I. Intimate Partner Violence Prevalence, Recurrence, Types, and Risk Factors Among Arab, and Jewish Immigrant and Nonimmigrant Women of Childbearing Age in Israel. J Interpers Violence. 2017:886260517705665).

Comment: Line 136: social and personal barriers. Please elaborate.

Answer: Personal and societal barriers refer to the following: personal barriers relate to health care providers (HCP) attitudes and perceptions regarding IPV; societal barriers are system-level barriers that relate to HCS and social services organized to provide professional support to women experiencing IPV. Please see p. 6, lines 153-155. 

Comment: Line 181: use (n=1401).

 Answer: Thank you. We included the word a cluster sample of 1401 (now line 200). 

Comment: Line 220: close spacing between study and ,

Answer: We have made this correction.

Comment: Study measures: well described. Salient and consistent with study intent.

Answer: Thank you.

Comment: Results: presented clearly. Adequate statistical tests and models selected.

Answer: Thank you. 

Comment, table 2. Line 309. May want to draw line after Arab women to highlight sample.

Answer: Thank you, we have done this in table 2. 

Comment: Line 315-316: use %'s.

Answer: Thank you, we have now add % to all (now lines 338-340). 

Comment: Again if sexual abuse was accounted for include results. Line 318-322

Answer: Yes, the physical IPV measure included sexual IPV. We have changed “physical IPV” to “physical and sexual IPV” across the paper. 

Comment: Multivariate associations drawn are accurate and consistent with research aims.

Answer: Thank you, 

Comment: Line 488: table 7 Include the n= in table heading to keep consistent with preservation of other tables.

Answer: Thank you, we have now included this in the Table 7 title. 

Comment: Discussion sections is well articulated as well.

Answer: Thank you 

Comment: Limitations are described and cited.

Answer: Thank you 

Comment: I would like to see the author(s) elaborate on the implications for practice and policy.

Answer: We have now added this to our discussion on pages 29-30 (lines 627-644). 

Comment: 

Again, I was to thank you for your valuable submission and encourage you to address the questions and comments. 

Regards –

Answer: Thank you very much.

---

## [Decision Letter · Decision Letter 1]

27 Nov 2019

PONE-D-19-17815R1

Patterns of healthcare services utilization associated with intimate partner violence (IPV): effects of IPV screening and receiving information on support services in a cohort of perinatal women

PLOS ONE

Dear Dr. Daoud,

Thank you for submitting your manuscript to PLOS ONE. After careful consideration, we feel that it has merit but does not fully meet PLOS ONE’s publication criteria as it currently stands. Therefore, we invite you to submit a revised version of the manuscript that addresses the last minor points raised during the review process as outlined below.

We would appreciate receiving your revised manuscript by Jan 11 2020 11:59PM. To enhance the reproducibility of your results, we recommend that if applicable you deposit your laboratory protocols in protocols.io, where a protocol can be assigned its own identifier (DOI) such that it can be cited independently in the future. For instructions see: http://journals.plos.org/plosone/s/submission-guidelines#loc-laboratory-protocols

We look forward to receiving your revised manuscript.

Kind regards,

Soraya Seedat

Academic Editor

PLOS ONE

Reviewers' comments:

Reviewer's Responses to Questions

**Comments to the Author**

1. If the authors have adequately addressed your comments raised in a previous round of review and you feel that this manuscript is now acceptable for publication, you may indicate that here to bypass the “Comments to the Author” section, enter your conflict of interest statement in the “Confidential to Editor” section, and submit your "Accept" recommendation.

Reviewer #1: All comments have been addressed

2. Is the manuscript technically sound, and do the data support the conclusions?

Reviewer #1: Yes

3. Has the statistical analysis been performed appropriately and rigorously? 

Reviewer #1: Yes

4. Have the authors made all data underlying the findings in their manuscript fully available?

Reviewer #1: Yes

5. Is the manuscript presented in an intelligible fashion and written in standard English?

Reviewer #1: Yes

6. Review Comments to the Author

Reviewer #1: Thank you for addressing the reviewers’ suggestions and re-submitting this very important manuscript. The clarity of the manuscript has been greatly improved with these edits. I have a couple of very minor suggestions below that would be helpful to address before publication (particularly point #4). Good work!

Minor Edits:

1) Page 3, Lines 71-74. This sentence needs to be restructured. I would suggest: “Women experience more IPV compared to men (5). As a result they are more likely to suffer the consequences of IPV including injury, sexual transmitted infections including HIV, depression, sleep and eating disorders, and alcohol and drug addictions.

2) Spell out abbreviations the first time they are introduced (e.g., EU on page 3, line 80) and then use them consistently without reintroduction (e.g., violence against women is not needed before VAW on page 4, line 101)

3) Page 4, line 98 seems to be missing a word: “Therefore a complex set of factors …”

4) Page 10, lines 260-262 – Should this be physical and/or sexual IPV? As well as emotional and/or verbal violence? I would assume that your factor analysis “clustered” these together based on their co-occurrence and that many women might be experiencing them together. If this is the correct interpretation, then it should be corrected throughout the manuscript.

7. PLOS authors have the option to publish the peer review history of their article (what does this mean?). If published, this will include your full peer review and any attached files.

Reviewer #1: No

---

## [Author Response · Author response to Decision Letter 1]

23 Dec 2019

PLOS ONE

Professor Soraya Seedat, 

Academic Editor

Dear Prof. Seedat, 

Thank you very much for your letter dated Nov. 27th informing us of the results of the review of our paper, “Patterns of healthcare services utilization associated with intimate partner violence (IPV): effects of IPV screening and receiving information on support services in a cohort of perinatal women” (PONE-D-19-17815).

We would like also to thank the reviewer for the positive evaluation of our paper and for suggesting minor revisions (see below). We have addressed each of these minor revisions and used track changes to highlight it in the original manuscript. 

Reviewer #1 comments: Thank you for addressing the reviewers’ suggestions and re-submitting this very important manuscript. The clarity of the manuscript has been greatly improved with these edits. I have a couple of very minor suggestions below that would be helpful to address before publication (particularly point #4). Good work!

Minor Edits:

1) Page 3, Lines 71-74. This sentence needs to be restructured. I would suggest: “Women experience more IPV compared to men (5). As a result they are more likely to suffer the consequences of IPV including injury, sexual transmitted infections including HIV, depression, sleep and eating disorders, and alcohol and drug addictions.

2) Spell out abbreviations the first time they are introduced (e.g., EU on page 3, line 80) and then use them consistently without reintroduction (e.g., violence against women is not needed before VAW on page 4, line 101)

3) Page 4, line 98 seems to be missing a word: “Therefore a complex set of factors …”

4) Page 10, lines 260-262 – Should this be physical and/or sexual IPV? As well as emotional and/or verbal violence? I would assume that your factor analysis “clustered” these together based on their co-occurrence and that many women might be experiencing them together. If this is the correct interpretation, then it should be corrected throughout the manuscript.

Best regards, 

Dr. Nihaya Daoud 

Corresponding author

---

## [Editor Report · Decision Letter 2]

8 Jan 2020

Patterns of healthcare services utilization associated with intimate partner violence (IPV): effects of IPV screening and receiving information on support services in a cohort of perinatal women

PONE-D-19-17815R2

Dear Dr. Daoud,

We are pleased to inform you that your manuscript has been judged scientifically suitable for publication and will be formally accepted for publication once it complies with all outstanding technical requirements.

With kind regards,

Soraya Seedat

Academic Editor

PLOS ONE
---

## [Editor Report · Acceptance letter]

13 Jan 2020

PONE-D-19-17815R2 

Patterns of healthcare services utilization associated with intimate partner violence (IPV): effects of IPV screening and receiving information on support services in a cohort of perinatal women 

Dear Dr. Daoud:

I am pleased to inform you that your manuscript has been deemed suitable for publication in PLOS ONE. Congratulations! Your manuscript is now with our production department. 

With kind regards,

on behalf of

Dr. Soraya Seedat 

Academic Editor

PLOS ONE